

# Spatiotemporal variations in the East Antarctic Ice Sheet during the Holocene

Takeshige Ishiwa[1,2], Jun'ichi Okuno[1,2], Yuki Tokuda[3], Satoshi Sasaki[4,*], Takuya Itaki[5], Yusuke Suganuma[1,2]

[1]Research Organization of Information and Systems, National Institute of Polar Research,10-3 Midori-cho, Tachikawa, Tokyo, 190-8518 Japan

[2]Polar Science Program, Graduate Institute for Advanced Studies, SOKENDAI, 10-3 Midori-cho, Tachikawa, Tokyo, 190-8518 Japan

[3]Faculty of Environmental Studies, Tottori University of Environmental Studies, 1-1 Wakabadai-kita, Tottori, Tottori, 689-1111 Japan

[4]Interdisciplinary Graduate School of Science and Engineering, Shimane University, 1060 Nishikawatsu-cho, Matsue, Shimane, 690-8504, Japan

[5]Paleogeodynamics Group, Geological Survey of Japan, National Institute of Advanced Industrial Science and Technology, 1-1-1 Umezoro, Tsukuba, Ibaraki, 305-8567 Japan

*Present address: Cooperative Faculty of Education, Gunma University, Aramakicho 4-2, Maebashi, Gunma, 371-8510 Japan

*Correspondence to*: Takeshige Ishiwa (ishiwa.takeshige@nipr.ac.jp)

**Abstract:** The past changes in East Antarctic Ice Sheet (EAIS) are crucial for understanding the ice sheet dynamics and its response to the Earth's climate system. Field-based geological data and various model simulations, such as ice sheet and glacial isostatic adjustment (GIA) modellings, provide significant insights into the behaviour of EAIS during the interglacial–glacial cycle. Recent in-situ cosmogenic nuclide surface exposure studies have revealed a large-scale thinning occurred in the Dronning Maud Land and Enderby Land of East Antarctica during 9–6 ka. However, the timing of this EAIS thinning event necessitates a revision of the ICE-6G model, which is a widely used GIA-based ice sheet history. To account for this temporal discrepancy, it is necessary to compare the sea levels calculated by GIA modelling with sea-level reconstructions to evaluate the validity of this refinement. The computed sea levels by GIA modelling are consistent with the relative sea-level reconstructions and indicate the spatial difference in the Holocene sea-level peaks, which is primarily due to the differences in the timings of ice-mass losses in the east and west of the Indian Ocean sector of East Antarctica. This finding challenges the prevailing assumption of synchronized ice-sheet growth and decay across this region, suggesting that the ice mass changes in the EAIS exhibit significant spatial differences.



## 1 Introduction

The Antarctic Ice Sheet (AIS) stores the largest volume of water on the Earth's surface, and its mass changes in the AIS have significantly influenced the global climate through ocean circulation and sea level changes. The East Antarctic Ice Sheet (EAIS) has an ice volume equivalent to the sea level of approximately 53 m (Fretwell et al., 2013), revealing its potential impact. Recent studies indicate that a part of the EAIS was lost compared with the present situation during the Last Interglacial under a climate about +1℃ warmer than the present (Crotti et al., 2022; Dutton et al., 2015; Iizuka et al., 2023; Wilson et al., 2018), highlighting the crucial importance of its stability in a warm future. However, despite the growing importance, investigating the spatiotemporal distribution of reconstructions is insufficient for quantifying ice mass changes and elucidating the mechanism of these changes (Jones et al., 2022). Notably, a comprehensive interpretation based on various modelling studies is essential for addressing these spatiotemporal gaps of reconstruction.

The glacial isostatic adjustment (GIA) modelling study plays an important role in reconstructing AIS changes (Argus et al., 2014; Briggs et al., 2014; Gomez et al., 2020; Ivins and James, 2005; Nakada and Lambeck, 1988; Whitehouse et al., 2012a). The GIA modelling utilizes the fact that the sea level approximates an equipotential surface of gravity to compute the response of the solid Earth to surface loading, considering the changes in seawater resulting from ice mass changes (Farrell and Clark, 1976). The ice loading history, which is one of the input values for the GIA model, can be constrained by comparing relative sea-level (RSL) reconstructions with the GIA model's computational results. For example, comparison with glacial RSL reconstructions can lead to the reconstruction of EAIS dynamics prior to the Last Glacial Maximum (Ishiwa et al., 2021a; Nakada et al., 2000), while comparison with Holocene RSL reconstructions can provide constraints on the timing of the deglaciation (Braddock et al., 2022).

Studies based on the RSL reconstructions of the Lützow–Holm Bay (LHB) in East Antarctica (Fig. 1) present extensive sea-level data on Antarctica (Miura et al., 1998a, b). These reconstructions provide significant insights into the history of the fluctuations in the EAIS during glacial periods (Ishiwa et al., 2021a; Nakada et al., 2000) to the Holocene (Verleyen et al., 2017). Detailed RSL reconstructions have been reported at important outcrops in Prydz Bay (PB), e.g. those of the Vestfold and Larsemann hills and Rauer Group (Berg et al., 2010a; Hodgson et al., 2009), which were used to reconstruct the EAIS history during the Holocene (Hodgson et al., 2016).



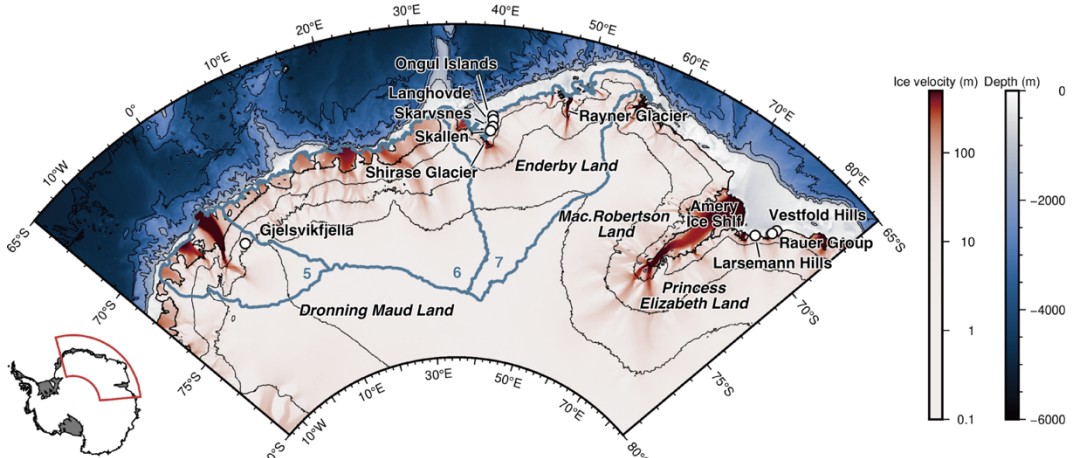

**Figure 1: Map of study sites. Ice velocity data are obtained from Rignot et al. (2017). Topography data are obtained from GEBCO2020 (GEBCO Bathymetric Compilation Group, 2019), and the contour interval is 1000 m. Thick blue lines indicate the Zwally Antarctic Drainage System 5–7. Figures in this study were developed using Generic Mapping Tool (Wessel et al., 2019).**

Surface exposure dating using cosmogenic nuclides is a method that estimates the duration for which the rocks have been exposed to cosmic rays by measuring the concentration of these nuclides (e.g., Gosse and Phillips, 2001; Nishiizumi et al., 1991). By dating the rocks that have been exposed due to ice retreat, we can determine the timing of the ice retreat. The cosmogenic nuclide dating of erratic and bedrock collected at various altitudes has been utilized to reconstruct the changes in heights of the EAIS since the Last Glacial Maximum(Andersen et al., 2023; Balco et al., 2023; Kawamata et al., 2020; Suganuma et al., 2014; Johnson et al., 2020; Suganuma et al., 2022; White et al., 2011; White and Fink, 2014; Yamane et al., 2011). Recent studies in Dronning Maud Land revealed early to mid Holocene ice-sheet thinning, indicating a discrepancy (delay) in the timing of deglaciation in previous studies that employed the ICE-6G model (Argus et al., 2014; Peltier et al., 2018). Suganuma et al. (2022) refined the ICE-6G model to fit the reconstruction of the field-based ice-sheet thinning that occurred from 9 ka to 5 ka, constrained by cosmogenic nuclide dates. However, the validity of this refinement was not assessed by comparing GIA-derived predictions with RSL reconstructions. This validation of the refined ice-loading history will improve the constraints on the ice-sheet changes in East Antarctica during the Holocene, thus, supporting highly accurate estimates of the GIA components, which is crucial for reducing the uncertainty in the present mass balance of the AIS. Therefore, in this study, we established a sea-level dataset for the LHB and PB regions, including the newly obtained data for the LHB, and assessed the validity of the refined ice-loading history using the established dataset and GIA modelling.



## 2 Methods

### 2.1 Glacial isostatic adjustment (GIA) model

The GIA model can be used to calculate the sea level changes while accounting for the solid Earth's deformation caused by surface loading changes (Farrell and Clark, 1976). In this study, we used a GIA model (Ishiwa et al., 2019, 2021b; Okuno and Nakada, 1999; Okuno et al., 2014; Suganuma et al., 2022) to predict the sea level for the study sites, while incorporating shoreline migration (Johnston, 1993), the gravitational attraction between the ice sheets and ocean (Nakada and Lambeck, 1989), and the Earth's rotational feedback (Milne and Mitrovica, 1998). There are spatial differences in rheology between East and West Antarctica, and studies on GIA using 3D models are advancing to understand their impact on the AIS dynamics (Pan et al., 2021). To address this issue, we set two kinds of rheology, "weak model" and "strong model", for our 1D GIA model. For the "weak model", we set the rheology for an elastic lithosphere thickness of 100 km, upper mantle viscosity of $5 \times 10^{20}$ Pa s, and lower mantle viscosity of $3 \times 10^{21}$ Pa s, as the VM5a parameter values (Argus et al., 2014; Peltier et al., 2015). For the "strong model", we set the rheology for an elastic lithosphere thickness of 100 km, upper mantle viscosity of $1 \times 10^{21}$ Pa s, and lower mantle viscosity of $3 \times 10^{21}$ Pa s (Whitehouse et al., 2012b).

The input topography for our GIA model was the ETOPO bedrock global relief model (Amante and Eakins, 2009; north of 60° S) and the BEDMAP2 bed elevation model (Fretwell et al., 2013; south of 60° S). The data were resampled to a resolution of 5 minutes, using The Generic Mapping Tools (Wessel et al., 2019). Combining the parameter of ice thickness in the ice-loading history in the GIA model with bedrock topography can produce more accurate results because this scheme can be used to reproduce ice shelves in the GIA calculation (Peltier et al., 2018; Purcell et al., 2016). In the ICE6G model used in this study (Argus et al., 2014; Peltier et al., 2015), the topography around Antarctica is based on BEDMAP2 (Fretwell et al., 2013). Consequently, the topography of the GIA model in this research adopts BEDMAP2. We think incorporating the latest topographic data, such as BEDMACHINE version 3 (Morlighem et al., 2020), would not affect the results of this study significantly due to the spatial resolution of our GIA model; the topography: 5 minutes, and the ice loading history is 15 minutes.

The ICE-6G_C (Argus et al., 2014; Peltier et al., 2015) and Nice6gSi6g_09-05_PART (Suganuma et al., 2022) models were introduced into our GIA model to reconstruct the ice-loading history over the past 122,000 years (Fig. 2). The Nice6gSi6g_09-05_PART model is a refined ICE-6G_C model based on the surface exposure dating results of Gjelsvikfjella and the Soya Coast in the Dronning Maud Land region (see Fig. 1) (Kawamata et al., 2020; Suganuma et al., 2022). In the Nice6gSi6g_09-05_PART model, the ice thicknesses in the Antarctic Drainage Systems 5–7 (covering the Dronning Maud Land and Enderby Land; Rignot et al., 2011) from 15 ka to 9 ka and from 6 ka to 0 ka are the same as those set at 15 ka and 0 ka, respectively. Fig. 2 portrays the spatial distribution of ice loading in the region at 9 ka, as estimated by the ICE-6G_C and Nice6gSi6g_09-05_PART model. The region of delayed deglaciation covers the RSL sites in the study area and the areas of Gjelsvikfjella, Skarvsnes, Skallen, Rayner Glacier that experienced ice-thinning during the mid-Holocene (Kawamata et al., 2020; Suganuma et al., 2022; White and Fink, 2014).



111

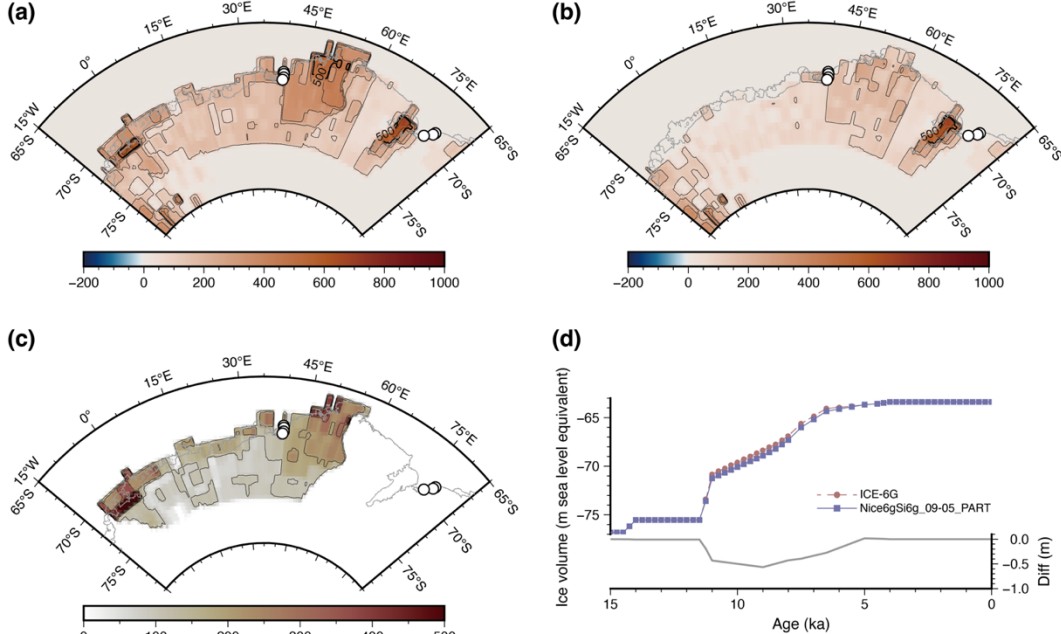

112

**Figure 2: (a–c) Circles indicate the relative sea-level (RSL) sites considered in this study. Difference in the present ice thickness and that at 9 ka using the (a) Nice6gSi6g_09–05_PART model and (b) ICE-6G_C model. (c) portrays the offset between (a) and (b). (d) Up: The red line denotes the volume change in the Antarctic Ice Sheet estimated using the ICE-6G_C model, and the blue line denotes the volume change estimated using the Nice6gSi6g_09–05_PART model. Bottom: The difference between the Nice6gSi6g_09–05_PART and ICE-6G_C models.**

118

## 2.2 Sea-level reconstructions

The RSLs are valuable indicator for constraining changes in AIS changes, and the interpretations of RSLs vary as marine or terrestrial limiting depending on the samples analysed (Briggs and Tarasov, 2013; Lecavalier et al., 2023; Shennan et al., 2015). The RSL records derived from shell fossils in raised beach sediments are indicative of marine limiting (Hayashi and Yoshida, 1994; Igarashi et al., 1995a, b; Maemoku et al., 1997; Miura et al., 1998a), while penguin remains suggest terrestrial limiting (Huang et al., 2009a, b, 2011). Furthermore, reconstruction of marine or lacustrine environments using isolation basin sediments provide evidence for marine and terrestrial limiting of RSLs respectively (Berg et al., 2010a, b; Hodgson et al., 2009, 2016; Takano et al., 2012; Verleyen et al., 2004, 2005, 2017). Our dataset was based on the compilations of previous RSL reconstructions of the LHB (Miura et al., 1998b) and PB (Hodgson et al., 2016) regions. In this study, we added the fossil shells (*Laternula elliptica* and *Adamusium colbecki*) collected during the geomorphological survey of the 61st Japanese Antarctic Research Expedition (e.g., Ishiwa et al., 2021a, 2022; Tamura et al., 2022) to the sea-level dataset of the LHB region. These shells maintain their living position and can be identified as *in situ*. Table 1 reveals the elevations of the samples corresponding to the sea-level values derived from the ellipsoid heights of the Reference Antarctic Elevation Model



(Howat et al., 2019), and geoid heights of EGM2008 (Pavlis et al., 2012), and mean dynamic ocean topography
(https://ftp.space.dtu.dk/pub/DTU10/; Andersen and Knudsen, 2009), determined using The Generic Mapping Tools (Wessel
et al., 2019). For trench samples (J61L-TrenchA-20-25, J61L-TrenchC-10-20, and J61L-TrenchC-28-34), the RSL values were
calculated from the samples' elevations and depth. The RSL values of other samples, which are surface sediments, correspond
to the elevations.
When discussing the dataset developed in this study, we excluded any data labelled as reworked in the previous work.
Additionally, the RSL reconstructions described as fragment were clearly marked on the figures and database due to the
possibility of redeposition. Where previous studies noted a range in elevation, this range was treated as a vertical uncertainty.
Otherwise, an uncertainty of ±1 m was assumed as in Lecavalier et al. (2023). The vertical error due to tide were set to ±0.8 m
(Aoyama et al., 2016; https://www.jodc.go.jp/vpage/tide.html) in LHB and ±0.9 m (Hodgson et al., 2016; Zwartz et al., 1998)
in PB, respectively. Furthermore, an additional error of ±1 m is added to consider paleo tides as in Briggs and Tarasov (2013).
The part of reported RSL reconstructions includes the radiocarbon ages that have not been adjusted for $\delta^{13}$C and
background corrections (e.g., Igarashi et al., 1995a, b; Miura et al., 1998b). Therefore, we used $\delta^{13}$C and background-corrected
radiocarbon ages for the LHB and PB sea-level compilation datasets. The radiocarbon ages in datasets were recalibrated using
the Oxcal software (Ramsey and Lee, 2013) with the Marine20 (Heaton et al., 2020) and SHCal20 (Hogg et al., 2020) curves.
For the LHB region, the local-reservoir age applied to the marine samples was set to 620±100 years (Verleyen et al., 2017;
Yoshida and Moriwaki, 1979); for the PB region, the age was set to 400±100 years (Hodgson et al., 2016), which was consistent
with the values compiled for the Southern Ocean (Berkman and Forman, 1996). We also recalibrated the age-depth models of
isolation basin sediment cores using the Bchron software (Haslett and Parnell, 2008). The sediments deposited in marine and
lacustrine environments were calibrated using the Marine 20 and SHCal20 curves.



**Table 1: Summary of the RSL reconstructions from the samples collected during the 61st Japanese Antarctic Research Expedition. The vertical error was set to ±0.8 m, which was identified by the tidal range in LHB (Aoyama et al., 2016; https://www.jodc.go.jp/vpage/tide.html). The calendar ages of radiocarbon dates were obtained using the Oxcal software (Ramsey and Lee, 2013) with the Marine20 (Heaton et al., 2020). The local reservoir was set to 620±100 years (Verleyen et al., 2017; Yoshida and Moriwaki, 1979).**

| Sample name | Region | Longitude (dd:mm:ss) | Latitude (dd:mm:ss) | Elevation (m) | Vertical error (±, m) | Materials | Lab. Code | $^{14}$C age (BP) | $\delta^{13}$C (‰) | Calendar age (cal BP) 2 sigma | Reference |
|---|---|---|---|---|---|---|---|---|---|---|---|
| J61L-0108-001 | Langhovde | -69:13.5205 | 39:39.7128 | 15.0 | 0.8 | *Laternula elliptica* | TKA-24127 | 6084 ±26 | -0.4±0.4 | 5385–5910 | This study |
| J61L-0110-001 | Langhovde | -69:13.509 | 39:39.743 | 4.3 | 0.8 | *Laternula elliptica* | TKA-24128 | 5812 ±25 | -0.5±0.2 | 5035–5600 | This study |
| J61L-0110-002 | Langhovde | -69:13.511 | 39:39.747 | 4.6 | 0.8 | *Laternula elliptica* | TKA-24129 | 5802 ±25 | -0.9±0.4 | 5029–5593 | This study |
| J61L-0110-003 | Langhovde | -69:13.51 | 39:39.749 | 4.4 | 0.8 | *Laternula elliptica* | TKA-24130 | 5844 ±26 | -1.7±0.3 | 5077–5645 | This study |
| J61L-0117-002 | Langhovde | -69:13.395 | 39:39.71 | 1.1 | 0.8 | *Laternula elliptica* | TKA-24134 | 2929 ±21 | 0.8±0.3 | 1470–2039 | This study |
| J61L-0118-005-Ra | Langhovde | -69:12.824 | 39:38.642 | 9.4 | 0.8 | *Laternula elliptica* | TKA-24135 | 6187 ±25 | 2±0.3 | 5480–6010 | This study |
| J61L-0118-006 | Langhovde | -69:12.7794 | 39:38.835 | 1.0 | 0.8 | *Laternula elliptica* | TKA-24136 | 4513 ±23 | 0.1±0.2 | 3393–3984 | This study |
| J61L-0118-008 | Langhovde | -69:12.779 | 39:39.765 | 1.1 | 0.8 | *Laternula elliptica* | TKA-24137 | 6700 ±26 | 0.5±0.3 | 6029–6580 | This study |
| J61L-TrenchA-20-25 | Langhovde | -69:13.508 | 39:39.746 | 4.3 | 0.8 | *Laternula elliptica* | TKA-24131 | 6518 ±27 | 0.3±0.3 | 5864–6378 | Tamura et al., 2022 |
| J61L-TrenchC-10-20 | Langhovde | -69:13.509 | 39:39.816 | 7.7 | 0.9 | *Laternula elliptica* | TKA-24132 | 6633 ±26 | -0.1±0.3 | 5956–6492 | Tamura et al., 2022 |
| J61L-TrenchC-28-34 | Langhovde | -69:13.509 | 39:39.816 | 7.7 | 0.9 | *Laternula elliptica* | TKA-24133 | 6793 ±28 | -1.2±0.4 | 6159–6683 | Tamura et al., 2022 |
| J61WO-0127-031 | Ongul Islands | -69:1.132 | 39:31.085 | 8.2 | 0.8 | *Adamusium colbecki* | TKA-24138 | 4163 ±23 | 1.2±0.3 | 2960–3541 | This study |
| J61WO-TrenchB-Surface | Ongul Islands | -69:1.132 | 39:31.085 | 8.2 | 0.8 | *Adamusium colbecki* | TKA-24139 | 4133 ±22 | 3.9±0.3 | 2918–3495 | This study |





## 3 Results

### 3.1 Relative sea-level (RSL) reconstructions in the Lützow–Holm Bay (LHB) and Prydz Bay (PB) regions

In the study area, terrestrial limiting data were indicated by penguin remains, and the lacustrine environment, defined with respect to the sea level not being higher than the sample's elevation or as the lowest sill around the isolation basin (Zwartz et al., 1998). Over the Holocene period, the sea levels of the Ongul Islands, Vestfold Hills, Rauer Group, and Larsemann Hills did not exceed 23, 8.8, 11.5, and 8 m, respectively; the sea level was constrained by the sediments from the isolation basin (Figs. 3 and 4). Note that RSL reconstructions at the Larsemann Hills show that the RSL exceeded 8 m in a short period based on Kirisjes Pond sediments (Hodgson et al., 2006; Verleyen et al., 2004, 2005).

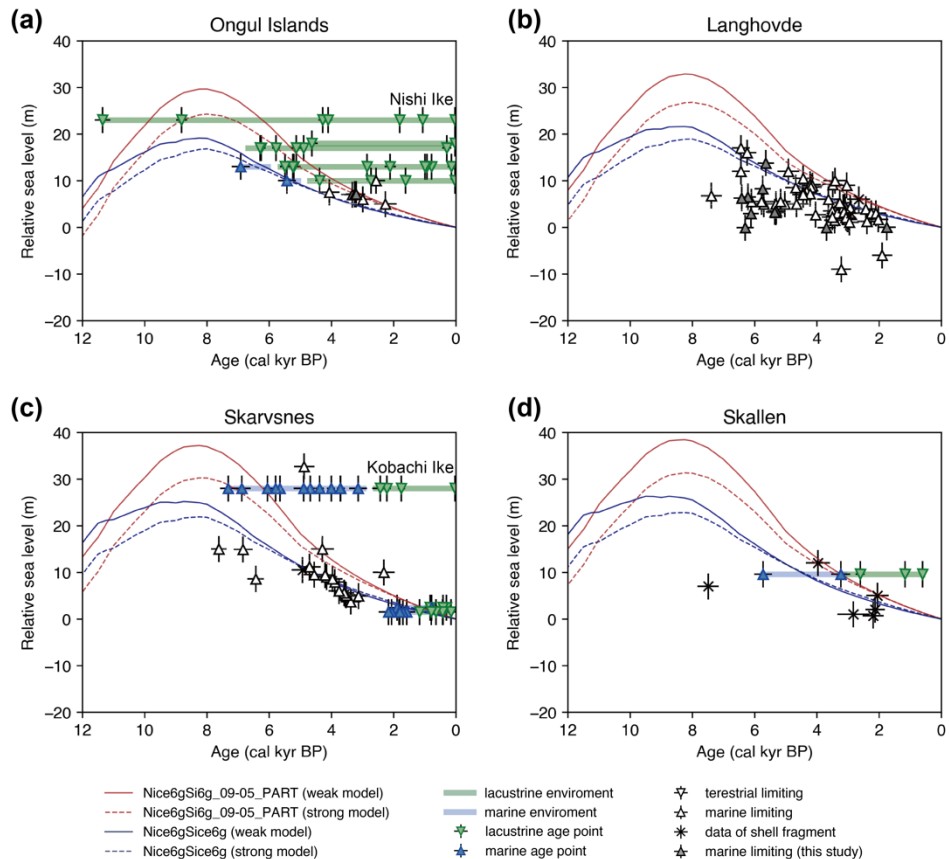

**Figure 3: Relative sea-level (RSL) reconstructions and the glacial isostatic adjustment (GIA)-predicted RSL of the Lützow–Holm Bay (LHB) over the past 12,000 years for (a) Ongul Islands, (b) Langhovde, (c) Skarvsnes, and (d) Skallen. Blue lines are the GIA-predicted RSLs using the ICE-6G_C model, and red lines denote the predictions carried out using the Nice6gSi6g_09-05_PART model. The solid lines denote the weak model (elastic lithosphere thickness of 100 km, upper mantle viscosity of $5 \times 10^{20}$ Pa s, and lower mantle viscosity of $3 \times 10^{21}$ Pa s). The dashed lines denote the strong model (elastic lithosphere thickness of 90 km, upper mantle viscosity of $1 \times 10^{21}$ Pa s, and lower mantle viscosity of $3 \times 10^{21}$ Pa s). The black upward pointed triangles denote the marine limiting of RSL reconstructions in this study. The white upward pointed triangles denote the previously reported marine limiting of RSL reconstructions. Crosses denote the data from the shell fragments. The blue upward pointed triangles and green downward**



pointed triangles indicate age points of the marine and lacustrine environments reconstructed by the isolation basin sediments, and
the blue and green thick lines represent durations of the marine and lacustrine environments, reconstructed by Bchron (Haslett and
Parnell, 2008). During lacustrine environments, sea level is below the sill height of the isolation basin, and during marine
environments, sea level is above the sill height of the isolation basin. Age uncertainty is two sigma.

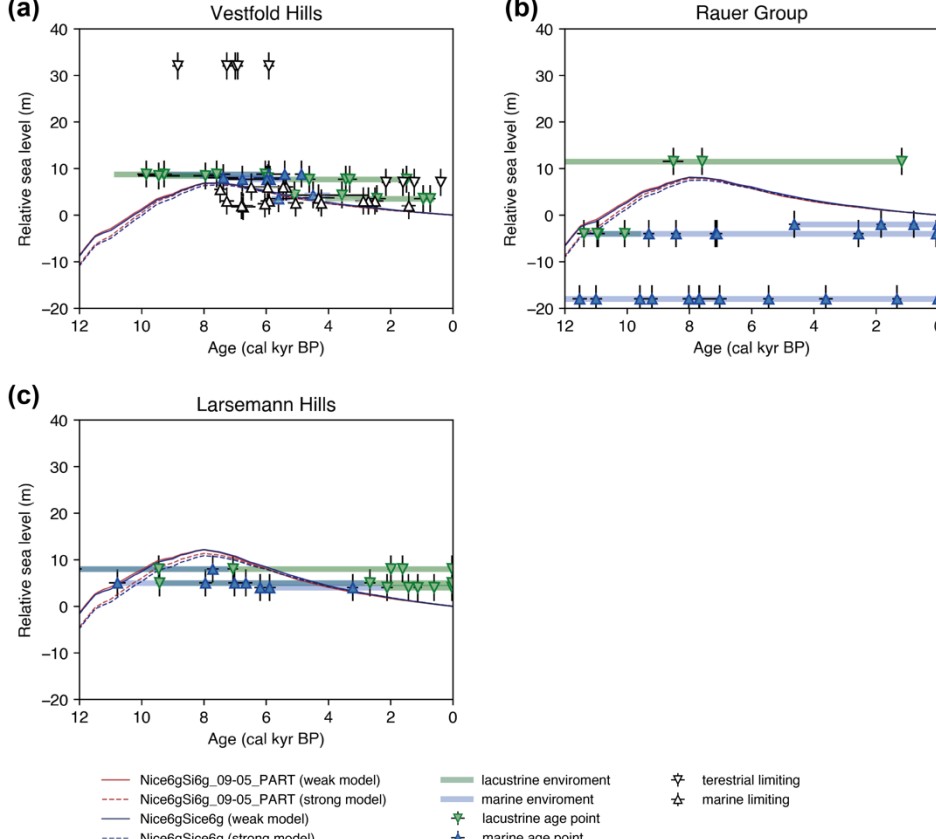


**Figure 4: Relative sea-level (RSL) reconstructions and the glacial isostatic adjustment (GIA)-predicted RSL of the Prydz Bay (PB)**
**over the past 12,000 years for (a) Vestfold Hills, (b) Rauer Group, and (c) Larsemann Hills. Blue lines are GIA-predicted RSLs using**
**the ICE-6G_C, and red lines denote the predictions carried out using the Nice6gSi6g_09-05_PART model. The solid lines denote the**
**weak model (elastic lithosphere thickness of 100 km, upper mantle viscosity of $5 \times 10^{20}$ Pa s, and lower mantle viscosity of $3 \times 10^{21}$**
**Pa s). The dashed lines denote the strong model (elastic lithosphere thickness of 90 km, upper mantle viscosity of $1 \times 10^{21}$ Pa s, and**
**lower mantle viscosity of $3 \times 10^{21}$ Pa s). The white upward pointed triangles and downward pointed triangles denote the previously**
**reported marine and terrestrial limiting of RSL reconstructions. The blue upward pointed triangles and green downward pointed**
**triangles indicate age points of the marine and lacustrine environments reconstructed by the isolation basin sediments, and the blue**
**and green thick lines represent durations of the marine and lacustrine environments, reconstructed by Bchron (Haslett and Parnell,**
**2008). During lacustrine environments, sea level is below the sill height of the isolation basin, and during marine environments, sea**
**level is above the sill height of the isolation basin. Age uncertainty is two sigma.**


195       The marine limiting data (indicated by shell fossils and marine environments) corroborated that the sea level was

higher than the elevation of the sampling site or that of the lowest sill around the isolation basin (Zwartz et al., 1998). Marine
limiting data were observed at all the LHB and PB sites, consistent with the RSL reconstructions that were based on terrestrial



limiting. If a marine limiting datum was available simultaneously with other marine limiting data, an RSL of the higher value
was adopted. The highest marine limiting values at Ongul Islands, Langhovde, Skarvsnes, and Skallen were 13 (at 6.9 cal kyr
BP), 17 (at 6.3 cal kyr BP), 33 (at 4.7 cal kyr BP), and 12 m (at 3.8 cal kyr BP) (Fig. 3). At Skarvsnes, the highest marine
limiting RSL, indicated by the shell fossils and the marine environments of Kobachi Ike (Verleyen et al., 2017), was delayed,
compared to other sites in the LHB region. The marine limiting data of the Ongul Islands, Langhovde, and Skarvsnes revealed
a trend of decreasing sea level; the data of the raised beach deposits in the region reflected sea-level regression. In Skallen, the
shell fragment shows the highest marine liming (Miura et al., 1998b), consistent with the marine limiting of 9.8 m by marine
environment in Lake Skallen (Fig. 3d).  The highest marine limiting values at Vestfold Hills, Rauer Group, and Larsemann
Hills were 5.8 (at 8.8 cal kyr BP), -2 (at 4.5 cal kyr BP), and 8 m (at 7.6 cal kyr BP). These data are from lakes and do not
reveal a similar trend of decreasing sea level, as that defined through the RSLs reconstructions of the LHB region (Figs. 3 and

208  4).

**3.2 Glacial isostatic adjustment (GIA) model output**
With respect to mid-Holocene peaks, the GIA-derived RSL predictions for the Nice6gSi6g_09-05_PART and ICE-6G_C
outputs for the weak model were 29.6 m and 18.9 m for the Ongul Islands, 32.8 m and 21.3 m for Langhovde, 37.2 m and 24.
8 m for Skarvsnes, 38.6 m and 24. 8 m for Skallen (Fig. 3), 6.4 m and 6.3 m for the Vestfold Hills, 7.6 m and 7.7 m for Rauer
Group, and 11. 9 m and 12.0 m for the Larsemann Hills (Fig. 4), respectively. In addition, the Nice6gSi6g_09-05_PART and
ICE-6G_C outputs for the strong model were 24.2 m and 16.7 m for the Ongul Islands, 26.8 m and 18.9 m for Langhovde,
30.4 m and 21. 9 m for Skarvsnes, 31.5 m and 22. 8 m for Skallen (Fig. 3), 6.3 m and 5.9 m for the Vestfold Hills, 7.4 m and
7.0 m for Rauer Group, and 11. 0 m and 10.6 m for the Larsemann Hills (Fig. 4), respectively. The general RSL trends for the
weak and strong models were similar at all sites. Notably, for the LHB region, the mid-Holocene RSL peaks of Nice6gSi6g_09-
05_PART were sharper than those of ICE-6G_C.

219       Fig. 5 shows the spatial distribution of the GIA-derived RSLs in the study area. A comparison of the Nice6gSi6g_09-

05_PART and ICE6G_C outputs revealed that the peak in the spatial distribution of the RSL in Nice6gSi6g_09-05_PART was
sharper and higher than that in ICE6G_C, with a difference of over 20 m in the weak model (Fig. 5e). Also, the weak model
portrayed sharper and higher RSL peaks for Nice6gSi6g_09-05_PART and ICE6G_C, compared to the strong model.

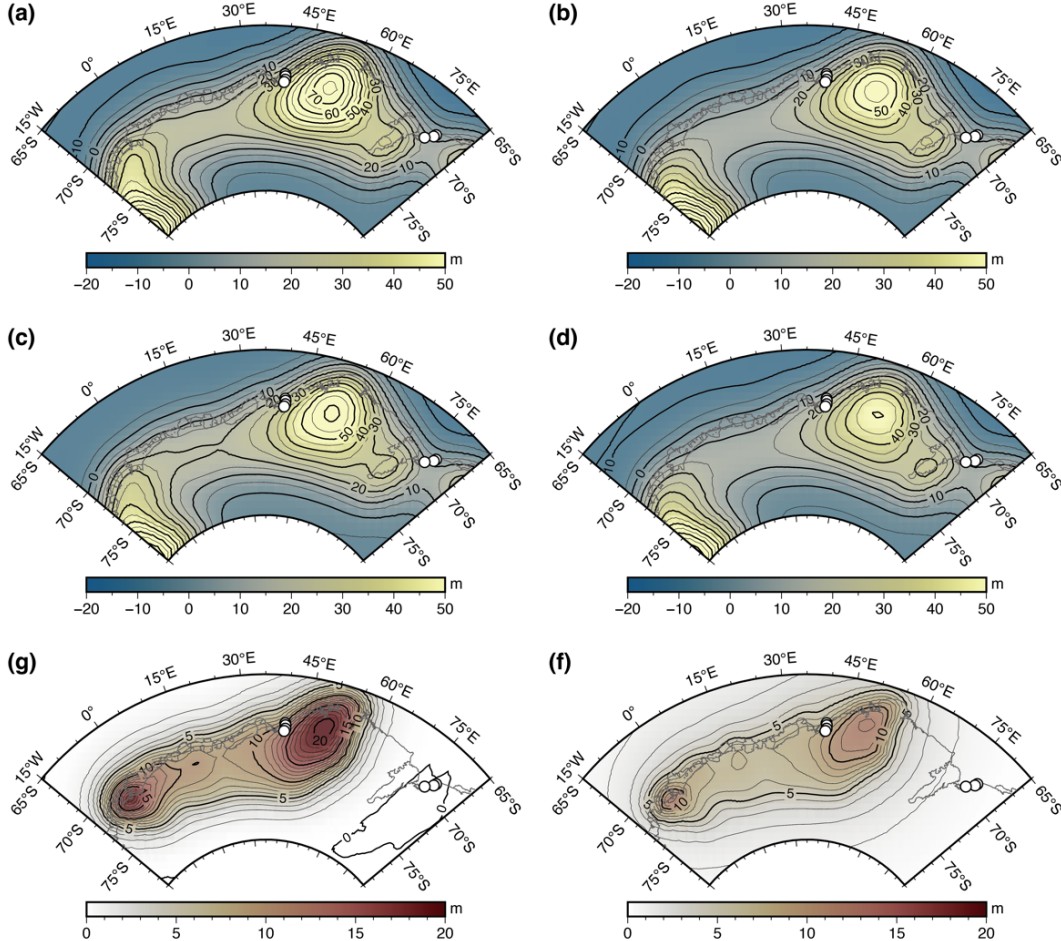

**Figure 5: Spatial distribution of relative sea-level (RSL) at 8 ka, based on the different ice-loading histories and rheology models used in this study. Circles indicate the discussed RSL sites for (a) Nice6gSi6g_09-05_PART and (b) ICE-6G_C for the weak model (elastic lithosphere thickness to 100 km, upper mantle viscosity to 5 × 10²⁰ Pa s, and lower mantle viscosity to 3 × 10²¹ Pa s). (c) Nice6gSi6g_09-05_PART and (d) ICE-6G_C outputs at 9 ka for strong model (elastic lithosphere thickness to 100 km, upper mantle viscosity to 1 × 10²¹ Pa s, and lower mantle viscosity to 3 × 10²¹ Pa s). (e) portrays the offset between (a) and (b). (f) presents the offset between (c) and (d).**

## 4 Discussion

A comparison of GIA model outputs with RSL reconstructions can reveal the changes in the EAIS; however, this requires an accurate assessment of sea-level uncertainty. The basic assumption is that the terrestrial and marine limiting obtained from geological archives indicate the upper and lower sea-level bounds, respectively. The marine limiting of RSL reconstructions in the LHB region can be dated to *Laternula ellipticus* and *Adamusium colbecki* (Miura et al., 1998). The reported habitat depth of *L. ellipticus* ranges from intertidal to approximately 700 m (Waller et al., 2017), and *A. colbecki* lives in shallow environments (Stockton, 1984). Because the reconstructions of the marine limiting from *L. ellipticus* and *A. colbecki* and the





terrestrial limiting (by lacustrine environments) were obtained from the strata of the age of ~2–4 cal kyr BP in the Ongul
Islands (Fig. 3a), we could corroborate the sea-level uncertainties by cross-referencing the two sets of records and conclude
that uncertainties of at least <5 m may be applicable for the Ongul Islands. The marine limiting for Skarvsnes were derived
from the samples of *L. ellipticus* and *A. colbecki* and fossilized worm tubes. However, the marine limiting of Kobachi Ike
portrayed uncertainties of >5 m, from ~3 cal kyr BP to 8 cal kyr BP (Fig. 3c). This indicates a regional difference in the sea-
level uncertainties between Skarvsnes and the Ongul Islands.
Surface exposure dating indicated a clear difference in the timing of ice-sheet thinning in the LHB and PB regions
(Kawamata et al., 2020; Suganuma et al., 2022; White et al., 2011; White and Fink, 2014). Skarvsnes and Skallen in the LHB
region experienced more than 400 m of ice-sheet thinning from 9 ka to 5 ka (Kawamata et al., 2020). Similar ice-sheet thinning
during the early–mid-Holocene was also observed in Gjelsvikfjella (Suganuma et al., 2022), Moreover, the surface-exposure
dating results of Rayner Glacier (Enderby Land; Fig. 1) revealed more than 400 m of ice-sheet thinning and more than 10 km
of ice retreat from 9 ka to 6 ka (White and Fink, 2014), suggesting that this event was a common phenomenon across the
Dronning Maud Land and Enderby Land regions.
In contrast, the surface exposure dating of the changes in the ice-sheet elevation in Macs. Robertson Land indicated
that at ~18 ka, ice-sheet thinning occurred downstream of the Lambert Glacier-Amery Ice Shelf system (LGAISS), reaching
the modern margin by ~12 ka. (White et al., 2011). In addition, the upstream area of the LGAISS experienced retreating ice
from 14 ka to 8 ka, portraying a delay when compared with the retreat noted downstream of the LGAISS, which may be due
to the time taken by the phenomenon to occur in the upstream area. By combining the surface-exposure dating results with the
weathering conditions and marine sediment records, White et al., (2022) concluded that the Raur Group and Vestfold Hills
became ice-free at ~15 ka, which could be associated with the grounding line retreat of the LGAISS. The ice sheets of Mac.
Robertson Land and Princess Elizabeth Land, including the LGAISS, are thought to have retreated at ~15 ka earlier than those
of Dronning Maud Land and Enderby Land.
The records of the changes in the ice-sheet elevation reconstructed from surface exposure dating indicated that the
timing of the reduction in the ice-sheet elevation varied at the boundary between Enderby Land and Mac. Robertson Land. We
referred to these findings when determining the regions for refining the ice-loading history based on ICE-6G_C (Fig. 2). The
influence of the refinement of ice-loading history on the reconstruction of global sea-level changes was not significant because
its contribution was less than 0.6 m (Fig. 2d). The RSL reconstructions and the results of GIA modelling by Nice6gSi6g_09-
05_PART were more consistent that these of ICE-6G_C, indicating that the regions selected in this study for refining the ice-
loading history were reasonable from the perspective of comparable RSL records.
Notably, the Nice6gSi6g_09-05_PART produced higher Holocene sea-level peaks than the ICE6G_C with the same
rheology (Figs. 3 and 4). The timing of the ice retreat of the Nice6gSi6g_09-05_PART was subsequent to the end of the global
sea-level rise mainly due to the ice-sheet retreat in the Northern Hemisphere (Lambeck et al., 2014). This temporal relationship
indicates that the global sea-level rise, which cancelled the local uplift by glacial rebound, was terminated before the glacial
uplift (with the beginning of the local ice retreat). Therefore, the uplift estimated in the Nice6gSi6g_09-05_PART model was





larger than that estimated in the ICE6G_C model with the same rheology, resulting in a higher sea-level highstand during the
Holocene.
The input of the rheology model properties into GIA modelling significantly influenced the GIA-derived RSLs.
Sensitivity tests were conducted using the weak and strong models. The GIA results obtained using the Nice6gSi6g_09-
05_PART and ICE6G_C models indicated that the weak model produced higher sea-level peaks during the Holocene for both
the LHB and PB regions (Figs. 3–5). This is because the weak model was more sensitive to the changes in loading than the
strong model. However, the differences in the GIA-derived RSLs between the weak and strong models were smaller for the
PB region than for the LHB region. This may be because local ice-sheet melting mostly terminated before the Holocene,
thereby minimizing the glacial isostasy effect.
The temporal distributions of the sea-level reconstructions for the LHB and PB regions also differed (Figs. 3 and 4).
The RSLs of the marine limiting based on the beach deposits and marine sediments in basins in Langhovde, Skarvsnes, and
Skallen (Fig. 3) were recorded only after 7 ka, indicating that these sites have been ice-free since at least 7 ka. This
interpretation is consistent with the reported timing of the ice-sheet thinning at Skarvsnes and Skallen using the surface
exposure ages (Kawamata et al., 2020). The marine limiting data covering the beginning of the Holocene in the Vestfold Hills
and Rauer Group (Fig. 4) were consistent with the timing of the ice-sheet retreat that was initiated in the LGAISS before the
Holocene (White et al., 2022). As this duration corresponds to global sea-level rise, mainly due to the ice-sheet retreat that
occurred in the Northern Hemisphere (Lambeck et al., 2014), we could conclude that the glacial rebound was cancelled by
local ice-sheet thinning (Hodgson et al., 2016), leading to a weak sea-level highstand in the PB region during the Holocene
(Fig. 4).
The inconsistency between the RSL reconstructions and the Nice6gSi6g_09-05_PART output for the Ongul Islands
(Fig. 3a) suggests a different local ice-sheet history within the LHB region. Nishi Ike on the West Ongul Island maintained
lacustrine conditions during the Late Holocene (Verleyen et al., 2017), indicating a terrestrial sea-level limiting of 23 m (Fig.
3). Hirakawa and Sawagaki (1998) reported that the highest elevation of a raised beach in the Ongul Islands was 20 m, lower
than the sea-level highstand of other exposed areas in the LHB region. The Nice6gSi6g_09-05_PART outputs for both the
strong and weak models exceeded this level. This indicates that the ice-loading history needs to be modified from the
perspective that a small amplitude of ice-loading or/and an earlier timing of ice retreat around the Ongul Islands may have
resulted in a small glacial rebound and a weak sea-level highstand during the Holocene. For Langhovde, the marine limiting
were more consistent with the Nice6gSi6g_09-05_PART outputs than the ICE-6G_C outputs. While the surface exposure ages
for Langhovde are yet to be reported, a compilation of GIA outputs and RSL reconstructions indicates that the timing of the
Holocene ice retreat synchronized with the retreats in Skarvsnes and Skallen, because the period estimated as "ice-free" by the
sea-level records of Langhovde matches the reported timings of ice-retreat in Skarvsnes and Skallen (Kawamata et al., 2020).
In Skarvsnes, the RSLs of the Nice6gSi6g_09-05_PART model are closer to the shell-fossil data and the marine
limiting data deduced from the marine environments of Kobachi Ike, compared with the RSLs of the ICE-6G_C model, with
the difference being significant (Fig. 3). To explain this discrepancy, further adjustments to the ice-loading history will be



needed, in addition to the corrections carried out in this study. Furthermore, a re-evaluation of the chronology or sedimentary
environment of the geological record will be necessary, including the re-evaluation of the values of the local reservoir for the
calibration of radiocarbon dating.

**Table 2: Summary of GIA-derived deformation vertical rates and the GNSS estimations by Hattori et al. (2021).**

| Site | Deformation vertical rates (mm/yr) | | | | GNSS estimations with the elastic deformation correction (Hattori et al., 2021) |
|---|---|---|---|---|---|
| | ICE6G with strong model | ICE6G with weak model | Nice6gSi6g_09-05_PART with strong model | Nice6gSi6g_09-05_PART with weak model | |
| Ongul Islands | 1.21 | 1.06 | 1.73 | 1.71 | 2.36±0.74 |
| Langhovde | 1.35 | 1.16 | 1.91 | 1.87 | 5.87±0.54 |
| Skarvsnes | 1.54 | 1.29 | 2.13 | 2.06 | 2.30±0.78 |
| Skallen | 1.59 | 1.33 | 2.20 | 2.11 | - |


In the LHB area, the GNSS observations have been conducted for about 30 years (Kazama et al., 2013; Ohzono et al.,
2006; Shibuya et al., 2003), and attempts were made to detect GIA signals from these observations (Hattori et al., 2021).
Hattori et al. (2021) indicate a discrepancy between the GIA signals results obtained through GNSS and the results of GIA
models, suggesting a need to discuss past ice sheet changes and the rheology. Table 2 shows the vertical deformation rates
calculated from GIA models, and regardless of the rheology adopted, the uplift rates for Nice6gSi6g_090-05_PART are
significantly higher compared to ICE-6G and more consistent with the estimations of GIA signals calculated from the GNSS
observations.
In the study area, we noted differences in the spatiotemporal distribution of ice loss and growth, suggesting that the
response mechanisms to loss and growth signals may differ by region. The area has been studied extensively and has a good
dataset of sea-level records for not only the Holocene period but also the MIS3. Using GIA modelling, (Ishiwa et al., 2021a)
explained why the MIS3 RSL reconstructions are higher than the present level. It was suggested that the ice-sheet volume from
Dronning Maud Land to Princess Elizabeth Land might have reached its maximum before the Last Glacial Maximum (~20,000
years ago; Ishiwa et al., 2019). We used RSL reconstructions and surface exposure ages to demonstrate that the timing of ice-
loss onset differed at the boundary between Enderby Land and Mac. Robertson Land. Thus, while ice-sheet growth occurs
synchronously across Dronning Maud Land and Princess Elizabeth Land (Ishiwa et al., 2021a), the timing of ice loss during
the glacial period varies by region, which is indicated by this study. To understand the factors behind the spatial differences in
ice sheet melting and growth, it is important to detect signals triggering ice sheet changes from the glacial to the Holocene in
marine sediment samples from these regions.



## 5 Conclusion

The obtained surface exposure dating by previous works indicates the occurrence of ice-sheet thinning in Dronning Maud Land and Enderby Land during the Holocene. The refined ICE-6G modelling carried out in this study (based on the surface exposure dating records) revealed higher sea-level peaks during the Holocene in the LHB, compared to the results of the original ICE-6G model. Notably, the GIA calculation results were consistent with the RSL reconstructions, indicating appropriate refinement. In contrast, Mac. Robertson Land and Princess Elizabeth Land experienced gradual ice retreats during the last deglaciation and Holocene. This earlier initiation of ice retreat did not result in the sea-level peaks in PB during the Holocene, which was consistent with the RSL reconstructions. The spatiotemporal differences in the sensitivity to the factors that drive the ice sheet changes contribute significantly to these spatial differences at the boundary between Enderby Land and Mac. Robertson Land. Thus, elucidating these differences can lead to detailed investigations pertaining to the response of ice-sheet variability to future climate-change conditions.

**Data availability:**

We have uploaded the sea-level dataset, age models of basin sediments, and glacial isostatic modelling results to Arctic and Antarctic Data archive System (https://ads.nipr.ac.jp/dataset/A20240131-001).

**Author contributions:**

TI carried out this research with the inputs from authors. JO supported GIA analysis. Shell samples were taken by JARE61 geomorphological survey members (TI, YT, SS, and TI). YS supervised this research and helped to write the manuscript. All authors approved this manuscript.

**Declaration of potential conflicts of interest:**

The authors declare that they have no conflict of interest.

**Acknowledgements:**

This study is a part of the Science Program of the Japanese Antarctic Research Expedition (JARE). It was supported by National Institute of Polar Research (NIPR) under MEXT. This study is funded by JSPS KAKENHI (21H01173 to TI, 19H00728 to YS, and 17H04983). We express thanks to Takayuki Omori for the radiocarbon measurement. Anonymous reviewers and the journal editor are greatly appreciated for their suggestions and recommendations. We would like to thank Editage (www.editage.com) for English language editing.



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
