# Peer review of "Spatiotemporal variations in the East Antarctic Ice Sheet during the"

_EGUsphere, 2024_

## Referee Comment (RC2)

**Review**

Ishiwa et al. 2024

**Content**

The authors model the RSL change during the Holocene in Dronning Maud Land and the surrounding applying two published glaciation histories and two earth structures. The histories are the original ICE6G_C model and a modified history based on surface exposure data, Nice6gSi6g_09-05_PART, in the following I call it Nice6g. The applied viscosity structures are the vm5a attached to ICE6g_C and one that was suggested by Whitehouse et al. (2012b) representing the Antarctic lithosphere and mantle. These the authors discuss as weak and strong, respectively.

The modelling was performed with a state-of-the-art GIA code, solving in addtion to the viscoelastic deformations the sea level equation, considering moving coast lines and rotational deformations.

The authors present new sea-level data, for the Lützow-Holm Bay covering the Holocene (9 kaBP to present day). They combine these data with further published compilations and compare them with the GIA model results at the the respective sites.

The conclusions based on this small set of GIA models are in general a better fit with Nice6g, but the results demand further improvements with respect to the glaciation history, where they argue for an asynchroneous change of the ice sheet in this region. For me it is not clear if such an asynchronicity is already considered in Nice6g. The authors also do not discuss wich of the two earth structures better fit.

Throughout the text, the authors focus on surface exposure data, although this is not the main topic of the paper. The authors apply the refined Nice6g model, which is based on this type of data. Although one of the authors of this paper is also author of Nice6g, you should reduce the discussion of this aspect and refer to the respective study.

**Language**

The authors used an AI tool to improve language. This explains to a large extent the style, which does not follow the conventions and nomenclature in the current scientific literature of the respective research fields. This makes the text sometimes tricky  to understand.

The text should really be improved by a native speaker which has experience in scientific writing. I recommend a more concise style of writing, as the authors tend to repeat statements at some places, which hinders the flow of the the text.

To show this I placed here some examples I took from the Introduction.

**Examples**

1. L33: 'The East Antarctic Ice Sheet (EAIS) has an ice volume equivalent to the sea level of approximately 53 m'
   I read as
   'The East Antarctic Ice Sheet (EAIS)  has an ice volume equivalent to a mean global baristatic sea level rise of approximately 53 m.'

2. L35: 'Recent studies indicate that a part of the EAIS was lost compared with the present situation during the Last Interglacial under a climate about +1°C warmer than the present'
   I read as
   'Recent studies indicate that during the Last Interglacial parts of the EAIS were lost compared to its current state, under a climate about +1°C warmer than today'.

3. L37: 'However, despite the growing importance, investigating the spatiotemporal distribution of reconstructions is insufficient for quantifying ice mass changes and elucidating the mechanism of these changes (Jones et al., 2022).'
   I read as
   'Jones et al. (2022), emphasised the need for a better understanding of the ice sheet dynamics in this region during the Holocene, which should be based on improved reconstructions of the spatiotemporal distribution of the EAIS.'
   The following sentence is redundant.

4. L41: 'The glacial isostatic adjustment (GIA) modelling study plays an important role'
   I read as
   'Modelling of glacial isostatic adjustment (GIA) is an important method to reconstruct'

**Scientific content**

I strongly recommend to extend the modelling in the direction the authors have stated in the outlook. My concerns regarding the language go hand in hand with the presentation of the results.

**Details**

**Specific phrases**

1. I would not call sea level data or index points being RSL reconstructions. From my understanding a reconstruction is based on an interpretation of a set of data points. So, reconstruction includes some modelling. In current literature these data are called sea level data points which are split into sea-level index points and limiting points.

2. **Nice6gSi6g_09–05_PART** I would abbreviate, as it reads a bit lengthy throughout the text.

3. RSL should not be used in plural form as it represents a measure. Also other terms like 'modelling' are not used in plural.

These are again only some examples.

**Text**

1. L25: Not clear, on which glaciation history the found consistency is based.

2. L 32: The authors state that the AIS significantly influenced global climate. Is there evidence for this and can you give a reference? A potential of 53 m sea level rise I would not rate a sufficient to explain its potential impact on the climate.

3. L41: The authors should also state that on the first place, GIA models the viscoelastic deformations in response to surface loading.

4. L 43: 'The GIA modelling utilizes the fact that the sea level approximates [...]'. This is strictly speaking not correct. In GIA modelling the sea level height is considered to follow the geoid. This is of course a good approximation, but the impact of ocean dynamics on the sea level, are neglected.

5. L50: LHB and PB are not indicated in Fig. 1.

6. L64: This sentence is not clear to me.

7. L 89: I would expect that the lithosphere thickness has a substantial impact, why do you keep it constant.

8. L92: Applying a software does not explain how the resampling is done. The authors should state, which method they applied.

9. L 93–95: The authors have described the method already above.

10. L 105: I guess they mean 'the drainage basins 5-7 according to the Antarctic Drainage System of Zwally et al. (2012).'

11. L106: I do not understand this description, are the 15 ka and 0 ka at the end of the sentence related to ICE6G_C?

12. L107: change to 'Figure 2', convention is to not to abbreviate 'Figure', when appearing at the beginning of a sentence.

13. L134: What are trench samples?

14. L151: Add 'respectively' at the end of the sentence.

15. L164: The sea level itself cannot be constrained.

16. L196-198: I read that in previous reconstructions only TL data are used, and that further ML data applied in this study confirm these findings.

17. L210–218: Why not list these data in a table. It is tricky to keep all these numbers in mind while reading and to have a clue on how the numbers differ between the respective regions.

18. L244–266: The authors discuss here in length the applied surface exposure dating, but this is not the scope of this study as the data is already applied in constructing Nice6g, and I assume they are also discussed in the respective referenced studies. I suggest to condense these three paragraphs and only summarise the findings regarding the adopted changes in the glaciation history.

19. L296ff: The authors state here that Nice5g needs further modifications, why not apply them? In the current study the changes to be me made read rather unspecific with many 'may be's'.

20. In addition to the adjustment of the glaciation history. What is the effect of the different earth structures? Which structure is the preferred one, if any?

21. L312–318: The authors present here additional data. This paragraph together with Table 2 should be placed in Section 3. Also, they only present here GNSS data for LHB. Are there no further data from the other regions discussed with respect to RSL?

22. L325ff: The authors should be more specific in which way the asynchronicity manifests.

23. L332ff: After reading the conclusions, the results seem to be rather preliminary. The first sentence is not a conclusion of this study as the glaciation history is already documented in the publication of Nice6g. In this section, I won't use past tence, as the results should stand.

24. L338: In this sentence, I am really not sure what the authors want to say.

**Figures**

In general the figure captions should be shortened.

Figure 1: The LHB is not indicated on the map. Is the Zwally Antarctic Drainage System a common phrase? In the text the authors only use Antarctic Drainage System. Also Zwally et al. 2012 should be cited then. The regions 5, 6 and 7, I would call 'drainage basins 5–7 following the Antarctic Drainage System of Zwally et al. (2012)',

The applied tool I would acknowledge in the acknowledgements. Furthermore, the figures were not developed but generated using GMT.

Figure 2: In d), the differences are really marginal. Also, Fig. 2 does not assist the discription, L105–107, especially how the different regions evolve over time.
I would change the sign of the unit, so that sea level equivalent represents a renormalised ice mass. Furthermore, why the authors do not plot the ice volume only for the regions of interest. Then, it would become more clear what happens. with the ice volume.

Table 1: 'vertical error was set to $\pm 0.8$ m' in the table as in the text also larger values appear. You should state the data only represent recent LHB data. You should add an appendix with the data presented in the HOLSEA format if you use new data.

Figure 3: I would not repeat here the parameterisations of the viscosity structures. The description of the different symbols used in the figure is really lengthy.

Figure 4: For details one can refer to Figure 3. This guaranties the reader, that the set up of the two figures is the same.

Figure 5: (g) should be (e).
Why not regroup, where the left column represents the results using the weak structure (a), (b), (e)=(a)-(b) and the right column the strong structure with (b), (d) and (f)=(b)-(d). The meaning of the circles I would explain at the end of the caption and also refere ato Fig 1.

**Conclusion**

My recommendation is between resubmission and major revision.

---

## Author Comment (AC1)

**Response to reviewer #1**

*We appreciate the reviewers for giving valuable comments and are pleased to resubmit this manuscript. The comments are in blue text and our responses in black and italic text.*

This paper presents new GIA modelling that uses two different ice loading histories (ICE6G and Nice6g...), alongside two values for mantle viscosity (weak and strong) to simulate spatiotemporal changes in RSL in the Indian Ocean facing sector of East Antarctica (Enderby Land and Mac Robertson Land). These modelled RSL histories are then compared to (mostly) previously published RSL reconstructions to assess which ice loading history and mantle viscosity are most consistent with the data. My understanding is that the Nice6g refinement was motivated by new cosmogenic data that shows a difference in deglaciation age from that applied in the ICE6G model. So to test if the refinement is appropriate this study uses it to model RSL.

In these terms this is a solid study but I find the main body of text quite difficult to follow if my understanding of the broader motivation is correct. The introduction would benefit from an expanded and explicit paragraph on the aims and objectives of this specific study. the last sentence (lines 75-76) doesn't fully articulate this.

*We have rewritten the last sentence of the introduction as 'Therefore, in this study, we established a sea-level dataset for the LHB and PB regions, including the newly obtained data for the LHB, and assessed the validity of the modified ice-loading history using the established dataset and the GIA modelling to identify the spatial variation in ice-mass changes in these regions.'*

I also think the paper slightly overstates its "sea level" reconstruction aspect. I appreciate the work that goes into recalibrating datasets so they are internally consistent and how this is done needs to be documented. But as far as I can tell the paper uses previously published RSL reconstructions supplemented by some new ages which, while i agree should be included, don't really change the RSL story. I think the paper would benefit from trimming down the RSL data side of things and being more focussed on, and expanding, the modelling aspect. It is, at heart, a modelling paper and should wear that badge with pride so-to-speak. One way to do this would be to remove "3.1 RSL reconstructions" from results and move it to a new section before the methods that covers "study sites".  This could describe both the deglaciation story (and the difference in timing relevant to ice loading histories) and the RSL curves (supplemented with new data).

*We have added the section of 'sea-level data' after introduction section.*

Similarly, the discussion covers a lot of material (i.e., lines 232-259) that, although relevant, would be better placed before results as it really sets the scene for the study (i.e., shows the different deglaciation timeframes) rather than being a point of discussion for the results of the work done here (i.e. GIA modelling). I then think the discussion needs a restructure, I'm not sure exactly how. Maybe discuss LHB and PB separately first before making comparisons. I think there are some interesting points here but they are quite hard to pull out from a discussion that jumps about so much.

I want to be supportive of this paper and think there is a body of work here that will be a good addition to our understanding of EAIS history and RSL change. However i think to realise its full potential and ensure it is picked up upon it needs some significant restructuring to more clearly state its motivation and focus more on the results of this study rather than re-discussing previous work.

*Thank you for the valuable comments. Based on your suggestion, we have changed the structure of the manuscript. By moving and removing the descriptions of surface exposure dates, we have focused more on the modelling aspects.*

**Specific comments**

Title: I don't think this title is appropriate. It doesn't really describe what this work did or what its contribution to knowledge is. "Refined ice loading histories improve fit of GIA models to relative sea level data in East Antarctica" is just one that comes to mind (I'm not a modeller so this maybe a poor suggestion). but i think something that better describes the work/conclusion of this paper is needed.

*Thank you for the suggestion. We have changed the title as "Spatial variation in Holocene sea-level change revealed by the timing of ice-mass loss in East Antarctica".*

Lines 50 - 55: Think the overall picture of RSL change needs to be described in the intro (or in a new section as suggested above)

*We have added these to the new section.*

Lines 62 - 67: Not needed, there is no new TCN data being presented so a description of the method in any form is not required.

*We have removed the sentences.*

Line 69: Ok so studies show difference in timing to that used in ICE6G...what did ICE6G use?

*The data from 42 GPS sites, 62 SEDs, 12 Holocene sea-level records, and 9 continental shelf sedimentary facies are used in ICE-6G (Argus et al., 2014), cited from Whitehouse et al. (2012). However, these datasets do not include SED data from Rayner Glacier (White and Fink, 2014), Gjelsvikfjella (Suganuma et al., 2022), and Soya Coast (Kawamata et al., 2020).*

Throughout the paper the authors refer to Nice6gSi6g_09-05_PART. This is really awkward to read inline and i wonder if it should be referred to more simply e.g., Suganuma ice history (vis a vis ICE6G).

*We have changed the name as mod-I6G_DML.*

My understanding is that the high marine limit on Skarvsnes is related to neotectonics but this isnt discussed anywhere. It has implications about the utility of the RSL record at this site for constraining GIA models. cf. discussion on lines 242-242.

*Thank you for the suggestion. We have added this to the discussion (L258).*

Lines 244-259 belong in an introduction or study site section.

*We have moved these sentences to the introduction.*

Fig 2. Units on colour bars missing. Coastline is really not clear. Label panels A/B with ice loading model.

*We have revised the figure as follows.*

[Figure]

*Figure 3: Ice loading at 9 ka using the (a) mod-I6G_DML model and (b) I6G model. (c) portrays the offset between (a) and (b). (d) Up: The red and blue lines denote the volume change in ADS 5-7 estimated using the I6G and mod-I6G_DML models. Bottom: The difference between these models. (a–c) Circles indicate the RSL sites considered in this study.*

Fig 3. Is Nice6gSice6g the same as ICE6G, check labels below panel C.

*We have revised the figure as follows.*

[Figure]

*Figure 2: RSL data and GIA-predicted RSL over the past 12,000 years for (a) Ongul Islands, (b) Langhovde, (c) Skarvsnes, (d) Skallen, (e) Vestfold Hills, (f) Rauer Group, and (g) Larsemann Hills. Blue and red lines are the GIA-predicted RSL using the I6G and mod-I6G_DML, respectively. Solid and dashed lines denote the weak and strong models of rheology, respectively. Black upward- pointed triangles denote marine limiting of RSL data in this study. White upward- and downward-pointed triangles denote previously reported marine and terrestrial limiting. Crosses denote the data from shell fragments. Blue upward- and green downward-pointed triangles indicate age points of marine and lacustrine environments obtained from isolation basin sediments, and blue and green thick lines represent durations of marine and lacustrine environments, established by Bchron (Haslett and Parnell, 2008). Age uncertainty is two sigma.*

Fig 5. Colour scheme should be divergent around zero surely? really hard to see at a glance where RSL is higher/lower than present. I would label the panels with the ice loading model/viscosity used. Its hard to keep referring back to the caption.

*We have revised the figure as follows.*

[Figure]

*Figure 4: Spatial distribution of relative sea-level (RSL) at 8 ka, based on the different ice-loading histories and rheology models used in this study. Circles indicate the discussed RSL sites for (a) I6G and (b) mod-I6G for the strong model. (d) I6G and (e) mod-I6G_DML outputs for weak model. (c) portrays the offset between (a) and (b). (f) presents the offset between (d) and (e).*

*We appreciate for your comments and look forward to your reply.*

*Sincerely,*

*Takeshige Ishiwa, Ph.D.*

---

## Author Comment (AC2)

**Response letter to reviewer #2**

*We appreciate the reviewers for giving valuable comments and are pleased to resubmit this manuscript. The comments are in blue text and our responses in black and italic text.*

**Major comments**

The authors model the RSL change during the Holocene in Dronning Maud Land and the surrounding applying two published glaciation histories and two earth structures. The histories are the original ICE6G_C model and a modified history based on surface exposure data, Nice6gSi6g_09-05_PART, in the following I call it Nice6g. The applied viscosity structures are the vm5a attached to ICE6g_C and one that was suggested by Whitehouse et al. (2012b) representing the Antarctic lithosphere and mantle. These the authors discuss as weak and strong, respectively.

The modelling was performed with a state-of-the-art GIA code, solving in addtion to the viscoelastic deformations the sea level equation, considering moving coast lines and rotational deformations.

The authors present new sea-level data, for the Lützow-Holm Bay covering the Holocene (9 ka BP to present day). They combine these data with further published compilations and compare them with the GIA model results at the respective sites.

The conclusions based on this small set of GIA models are in general a better fit with Nice6g, but the results demand further improvements with respect to the glaciation history, where they argue for an asynchroneous change of the ice sheet in this region. For me it is not clear if such an asynchronicity is already considered in Nice6g. The authors also do not discuss witch of the two earth structures better fit.

Throughout the text, the authors focus on surface exposure data, although this is not the main topic of the paper. The authors apply the refined Nice6g model, which is based on this type of data. Although one of the authors of this paper is also author of Nice6g, you should reduce the discussion of this aspect and refer to the respective study.

*Thank you for your suggestion. We have reduced the description of surface exposure data and focused more on the GIA model. The section of sea-level data has added before the Methods section. The data from 42 GPS sites, 62 SEDs, 12 Holocene sea-level records, and 9 continental shelf sedimentary facies are used in ICE-6G (Argus et al., 2014), cited from Whitehouse et al. (2012). However, these datasets do not include SED data from Rayner Glacier (White and Fink, 2014), Gjelsvikfjella (Suganuma et al., 2022), and Soya Coast (Kawamata et al., 2020).*

**Language**

The authors used an AI tool to improve language. This explains to a large extent the style, which does not follow the conventions and nomenclature in the current scientific literature of the respective research fields. This makes the text sometimes tricky to understand.

The text should really be improved by a native speaker which has experience in scientific writing. I recommend a more concise style of writing, as the authors tend to repeat statements at some places, which hinders the flow of the the text. To show this I placed here some examples I took from the Introduction.

*Thank you for your comments. We have revised the manuscripts based on the comments from the English editing service, Editage.*

**Examples**

1.  L33: 'The East Antarctic Ice Sheet (EAIS) has an ice volume equivalent to the sea level of approximately 53 m'

I read as 'The East Antarctic Ice Sheet (EAIS) has an ice volume equivalent to a mean global baristatic sea level rise of approximately 53 m.'

*We have revised the sentence.*

2. L35: 'Recent studies indicate that a part of the EAIS was lost compared with the present situation during the Last Interglacial under a climate about +1°C warmer than the present'
I read as 'Recent studies indicate that during the Last Interglacial parts of the EAIS were lost compared to its current state, under a climate about +1°C warmer than today'.

*We have revised the sentence.*

3. L37: 'However, despite the growing importance, investigating the spatiotemporal distribution of reconstructions is insufficient for quantifying ice mass changes and elucidating the mechanism of these changes (Jones et al., 2022).'
I read as 'Jones et al. (2022), emphasised the need for a better understanding of the ice sheet dynamics in this region during the Holocene, which should be based on improved reconstructions of the spatiotemporal distribution of the EAIS.' The following sentence is redundant.

*We have revised the sentence and removed the following sentence.*

4. L41: 'The glacial isostatic adjustment (GIA) modelling study plays an important role' I read as 'Modelling of glacial isostatic adjustment (GIA) is an important method to reconstruct'

*We have revised the sentence.*

**Scientific content**

I strongly recommend to extend the modelling in the direction the authors have stated in the outlook. My concerns regarding the language go hand in hand with the presentation of the results.
We have changed the structure of the manuscript

*By adding the sea-level data section before the Methods, the structure of the manuscript has been adjusted to focus more on the modelling aspect.*

**Details Specific phrases**

1. I would not call sea level data or index points being RSL reconstructions. From my understanding a reconstruction is based on an interpretation of a set of data points. So, reconstruction includes some modelling. In current literature these data are called sea level data points which are split into sea-level index points and limiting points.

*We have replaced reconstructions with data.*

2. **Nice6gSi6g_09–05_PART** I would abbreviate, as it reads a bit lengthy throughout the text.

*We have renamed the model as mod-I6G_DML.*

3. RSL should not be used in plural form as it represents a measure. Also other terms like 'modelling' are not used in plural. These are again only some examples.

*We have revised the words.*

**Text**

1.  L25: Not clear, on which glaciation history the found consistency is based.

    *We have revised the sentence.*

2.  L 32: The authors state that the AIS significantly influenced global climate. Is there evidence for this and can you give a reference? A potential of 53 m sea level rise I would not rate a sufficient to explain its potential impact on the climate.

    *We have moved the reference.*

3.  L41: The authors should also state that on the first place, GIA models the viscoelastic deformations in response to surface loading.

    *We have revised the sentence.*

4.  L 43: 'The GIA modelling utilizes the fact that the sea level approximates [...]'. This is strictly speaking not correct. In GIA modelling the sea level height is considered to follow the geoid. This is of course a good approximation, but the impact of ocean dynamics on the sea level, are neglected.

    *We have revised the sentence.*

5.  L50: LHB and PB are not indicated in Fig. 1.

    *We have revised the figure.*

6.  L64: This sentence is not clear to me.

    *We have removed this part.*

7.  L 89: I would expect that the lithosphere thickness has a substantial impact, why do you keep it constant.

    *Thank you for your comment. Comparison with data has reported that lithosphere thickness does not influence sensitivity to RSL (Whitehouse et al., 2012). Additionally, we conducted a sensitivity test for lithosphere thickness, suggesting that the results showed little variation. Therefore, this study adopted a lithosphere thickness of 100 km.*

[Figure]

*Figure R2-1: The results of sensitivity tests of lithosphere thickness. Solid lines are upper mantle viscosity of 5 × 10²⁰ Pa*

*s, and lower mantle viscosity of $3 \times 10^{21}$ Pa s. Dashed lines are upper mantle viscosity of $1 \times 10^{21}$ Pa s, and lower mantle viscosity of $3 \times 10^{21}$ Pa s.*

8. L92: Applying a software does not explain how the resampling is done. The authors should state, which method they applied.

*We have revised the sentence.*

9. L 93–95: The authors have described the method already above.

*We have removed this sentence.*

10. L105: I guess they mean 'the drainage basins 5-7 according to the Antarctic Drainage System of Zwally et al. (2012).'

*Thank you for your suggestion. There was mistake of citation and we have used the Drainage System from Rignot et al. (2011). https://nsidc.org/data/NSIDC-0709*

11. L106: I do not understand this description, are the 15 ka and 0 ka at the end of the sentence related to ICE6G_C?

*Yes, they are related to ICE6G_C. We have revised the sentence.*

12. L107: change to 'Figure 2', convention is to not to abbreviate 'Figure', when appearing at the beginning of a sentence.

*We have revised the word.*

13. L134: What are trench samples?

*We have added the explanation to L144.*

14. L151: Add 'respectively' at the end of the sentence.

*We have added the word.*

15. L164: The sea level itself cannot be constrained.

*We have revised the word.*

16. L196-198: I read that in previous reconstructions only TL data are used, and that further ML data applied in this study confirm these findings.

*We have removed this sentence since it leads to misunderstanding.*

17. L210–218: Why not list these data in a table. It is tricky to keep all these numbers in mind while reading and to have a clue on how the numbers differ between the respective regions.

*We have made the table 2, summarising the Holocene sea-level peaks in study sites.*

18. L244–266: The authors discuss here in length the applied surface exposure dating, but this is not the scope of this study as the data is already applied in constructing Nice6g, and I assume they are also discussed in the respecitive referenced studies. I suggest to condense these three paragraphs and only summarise the findings regarding the adopted changes in the glaciation history.

*We have moved these sentences to an introduction.*

19. L296ff: The authors state here that Nice5g needs further modifications, why not apply them? In the current study the changes to be me made read rather unspecific with many 'may be 's'.
*We have revised the sentences.*

20. In addition to the adjustment of the glaciation history. What is the effect to f the different earth structures? Which structure is the preferred one, if any?
*We have discussed about the rheology in L231–L237. It is difficult to determine which is preferred because there is the trade-off between ice loading and rheology.*

21. L312–318: The authors present here additional data. This paragraph to gether with Table 2 should be placed in Section 3. Also, they only present here GNSS data for LHB. Are there no further data from the other regions discussed with respect to RSL?
*We have moved Table 2 to Results section and have added to the GNSS estimation for PB.*

22. L325ff: The authors should be more specific in which way the asynchronicity manifests.
*We have revised the sentence.*

23. L332ff: After reading the conclusions, the results seem to be rather preliminary. The first sentence is not a conclusion of this study as the glaciation history is already documented in the publication of Nice6g. In this section, I won't use past tence, as the results should stand.
*Thank you for suggestion. We have revised the sentence.*

24. L338: In this sentence, I am really not sure what the authors want to say.
*We have removed the sentence.*

**Figures**

In general the figure captions should be shortened.
*We have rewritten the captions.*

Figure 1: The LHB is not indicated on the map. Is the Zwally Antarctic Drainage System a common phrase? In the text the authors only use Antarctic Drainage System. Also Zwally et al. 2012 should be cited then. The regions 5, 6 and 7, I would call 'drainage basins 5–7 following the Antarctic Drainage System of Zwally et al. (2012)',
*Thank you for the suggestion. We have mistaken the citation of used drainage. We have used the drainage system from Rignot et al. (2011).*

The applied tool I would acknowledge in the acknowledgements. Furthermore, the figures were not developed but generated using GMT.
*Thank you for the suggestion. We have revised the sentence.*

**Figure 2:** In d), the differences are really marginal. Also, Fig. 2 does not assist the discription, L105–107, especially how the different regions evolve over time. I would change the sign of the unit, so that sea level equivalent represents a renormalised ice mass. Furthermore, why the authors do not plot the ice volume only for the regions of interest. Then, it would become more clear what happens. with the ice volume.

*Thank you for the suggestion. We have revised the figure to focus only on the region. The unit was set to sea level equivalent to support discussions about the relationship with the global sea-level.*

**Table 1:** 'vertical error was set to m' in the table as in the text also larger values appear. You should state the data only represent recent LHB data. You should add an appendix with the data presented in the HOLSEA format if you use new data.

Thank you for your suggestion. We have revised the vertical uncertainty in the table. We have uploaded the sea level data used in this study to ADS (https://ads.nipr.ac.jp/dataset/A20240131-001), which meets the criteria of HOLSEA.

**Figure 3:** I would not repeat here the parameterisations of the viscosity structures. The description of the different symbols used in the figure is really lengthy.

*We have revised the captions.*

**Figure 4:** For details one can refer to Figure 3. This guaranties the reader, that the set up of the two figures is the same.

*We have merged Figure 3 and 4 to Figure 2.*

[Figure]

*Figure 2: RSL data and GIA-predicted RSL over the past 12,000 years for (a) Ongul Islands, (b) Langhovde, (c) Skarvsnes, (d) Skallen, (e) Vestfold Hills, (f) Rauer Group, and (g) Larsemann Hills. Blue and red lines are the GIA-predicted RSL using the I6G and mod-I6G_DML, respectively. Solid and dashed lines denote the weak and strong models of rheology, respectively. Black upward- pointed triangles denote marine limiting of RSL data in this study. White upward- and downward-pointed triangles denote previously reported marine and terrestrial limiting. Crosses denote the data from shell fragments. Blue upward- and green downward-pointed triangles indicate age points of marine and lacustrine environments*

*obtained from isolation basin sediments, and blue and green thick lines represent durations of marine and lacustrine environments, established by Bchron (Haslett and Parnell, 2008). Age uncertainty is two sigma.*

Figure 5: (g) should be (e).

Why not regroup, where the left column represents the results using the weak structure (a), (b), (e)=(a)-(b) and the right column the strong structure with (b), (d) and (f)=(b)-(d). The meaning of the circles I would explain at the end of the caption and also refere ato Fig 1.

*We have revised the figure as follows.*

[Figure]

*Figure 4: Spatial distribution of relative sea-level (RSL) at 8 ka, based on the different ice-loading histories and rheology models used in this study. Circles indicate the discussed RSL sites for (a) I6G and (b) mod-I6G for the strong model. (d) I6G and (e) mod-I6G_DML outputs for weak model. (c) portrays the offset between (a) and (b). (f) presents the offset between (d) and (e).*

**Conclusion**

My recommendation is between resubmission and major revision.

*We appreciate for your comments and look forward to your reply.*

*Sincerely,*

*Takeshige Ishiwa, Ph.D.*